# Human Aligned Reward Modeling for Automated Transfer Function Generation of 3D Rendering of Medical Image Data

Anonymous Full Paper
Submission 15

## Abstract

In recent years, the quality of medical image data, such as computed tomography or magnetic resonance tomography, has continued to improve and the resolution and detection of the smallest structures has become increasingly accurate. Along with these developments, new techniques for three-dimensional visualization using volume rendering techniques are emerging, enabling extremely realistic visualization of medical images. This helps to improve patient communication, diagnosis, and treatment planning. An extremely critical step in the development of a realistic rendering is the design of a suitable transfer function. However, this requires a high level of experience and manual fine-tuning to the given image data. To automatize this process, we propose to train a reinforcement learning agent that extracts a two-dimensional transfer function from the given joint histograms of the image data. The focus of this study is primarily on the development of a suitable reward model, which is critical for the reinforcement learning framework, incorporating human feedback.

## 1 Introduction

Today, medical image data, with its extremely high quality, are playing an increasingly crucial part in the diagnosis and treatment planning of various diseases. Volumetric data sets can be acquired from different imaging modalities, such as computed tomography (CT), magnetic resonance imaging (MRI), positron emission tomography (PET), or ultrasound (US). Particularly, CT and MRI offer high image resolution with a high amount of anatomical details. These high-quality medical images allow the creation of renderings with exceptional detail and realism. First 3D visualizations of different structures, such as bones or organs, were obtained using iso-surface extraction based on previous segmentation. However, these methods, known as indirect volume rendering (iDVR), have the disadvantage that it is difficult to differentiate between adjacent structures based on a singular isovalue. Hence, many small sub-volumes, including the neighboring anatomical surroundings, would have to be segmented and visualized in order to represent adjacent structures. On the other hand, direct volume rendering (DVR) offers a flexible and detailed representation of volume data based on the direct mapping of the entire data volume without prior surface extraction using a transfer function (TF). The design of the TF plays a crucial role in generating DVR images with high quality and focusing on different regions of interest. An interactive design of the TF allows the user to define which anatomical structures are to be emphasized and how they are displayed in the subsequent DVR image in terms of their optical properties (opacity and color). However, this manual TF design is often not intuitive, repetitive, and time-consuming [1]. This is due to the fact that this design process is implemented on an intermediate level in two-dimensional feature spaces representing certain image characteristics, such as intensity and gradients, in the so-called joint histogram (JH). Selecting suitable features and the subsequent extraction of the TF are not trivial and require a high level of experience. In addition, a manual design of the TF requires adaptation to new image data and different visualization scenarios. To simplify this design process, alternative iterative procedures were developed, starting from an initial TF, to improve it towards an optimal solution which satisfies a pre-defined objective metric [2–4]. However, these approaches were only capable of optimizing regions along the same ray and could not include neighboring information from other rays. Further, the TF parameters to be optimized were concentrated only on opacity. The color, which is also necessary to determine the visual attention of different anatomical structures was not considered [5]. Another disadvantage of defining a dedicated optimization metric is that it is often non-trivial, and a good visualization result is difficult for humans to achieve based on certain defined mathematical properties. Many approaches that use learning based techniques, such as CNNs, to automate the rendering design have the same difficulties. In addition, labeled data are usually required to train the networks [6]. As a result, new methods are becoming increasingly popular that make the design of an objective function based on predefined criteria obsolete. One such approach is reinforcement learning from human feedback (RLHF), which has gained increased attention in recent years. Instead of directly formulating an objective function, RLHF uses collections of preferences provided by a human judge or inspector to train an RL agent, as suggested by Christiano et al. [7]. We want to

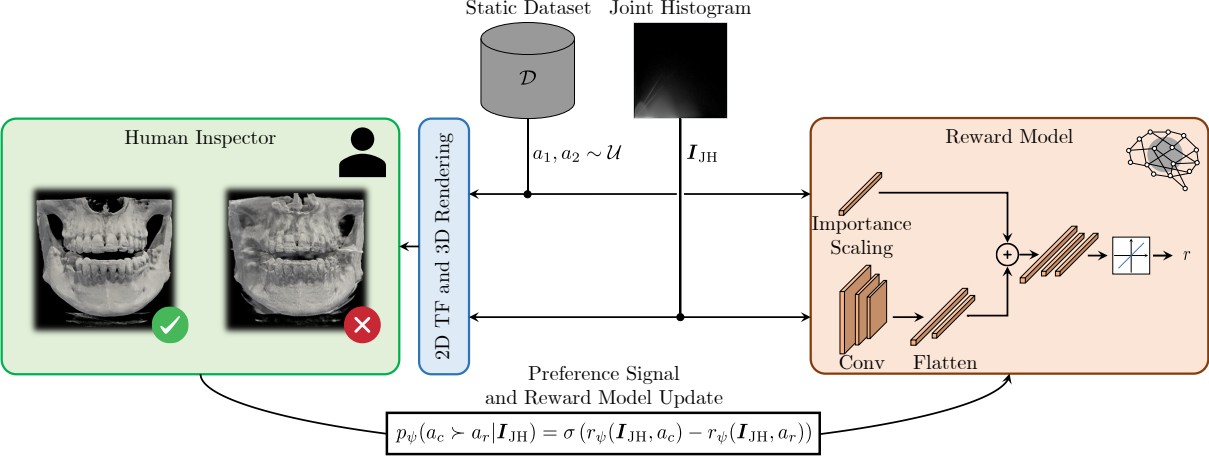

**Figure 1.** Schematic representation of the offline reward model training concept for an RL framework to automatically generate 2D TFs. The same JH, which is previously calculated from the image data, and two randomly drawn actions, in our case defined by vertices of a polygon, are used to calculate two 3D renderings, which will be presented to a human inspector. The reward model receives as input the JH, which is passed through three convolution layers each with a dropout probability of $p = 50\%$ to extract the image features. These are then concatenated with the up-scaled features of the actions, which are processed together and finally classified by a linear activation to predict a scalar reward. The reward model update is performed based on the preference collection of the human inspector rating the two generated 3D renderings.

adopt this approach here as well, to generate an automated 2D TF with the help of an RL agent, which is trained based on user feedback on the generated volume renderings from the learned TF.

## 2 Methods

We follow the RLHF pipeline proposed by Ziegler et al. [8], which typically includes three phases: the supervised fine-tuning of an agent, the preference collection for a subsequent reward model training, and the RL fine-tuning using proximal policy optimization (PPO) [9]. In this work, we focus on the second stage of this pipeline and present a suitable reward model, which is critical for the successful training of the RL agent.

In our RL framework, we define the state of the environment to

$$s = \boldsymbol{I}_{\mathrm{JH}}, \tag{1}$$

with $\boldsymbol{I}_{\mathrm{JH}}$ as the previously calculated image of the JH. It represents the intensities and gradients of the original 2D slices of the input images. The task of the agent is now to estimate suitable vertices of a polygon in this JH, which is used to calculate the 2D TF, in order to align the resulting 3D rendering with the visual imagination of the human inspector. The agent's action can thus be formulated as

$$a = [(x_1, y_1), (x_2, y_2), \ldots, (x_n, y_n)]^{\mathrm{T}}, \tag{2}$$

with $x$ and $y$ being the coordinates of the polygon's vertices inside the JH. The RL framework is implemented as a one-shot learning method allowing the agent to find the vertices of the polygon in one time step per episode. Therefore the polygon must be as representative as possible for the later representation of the desired anatomical features with the corresponding color and opacity values. In order to keep the possible action space as simple and small as possible with regard to the degree of freedom, the agent's initial task is to determine four corner points. However, the number of points can be increased after the first successful application to enable an even more precise TF definitions. Finally, the reward $r$ for the agent is provided by the reward model, which indicates the quality of the action given the state.

Figure 1 shows the general structure for the training as well as the reward model's architecture. For the offline training of the reward model we exclude the agent. However, we provide pre-defined human labeled and random generated actions, which are stored in a static dataset $\mathcal{D}$. The reward model receives as input the image of the JH together with an action representing the four vertices of the polygon. The image of the JH is propagated through three convolutional layers each with a dropout probability of $p = 50\%$ to extract the image features. These are then concatenated with the up-scaled features of the action in order to maintain a balance between the action and images features. In the last layers, both the image and the action features are processed together and finally classified by a linear activation to predict the scalar reward $r$. The update of the reward model is performed based on the preference signal by a human inspector rating two 3D renderings. This is achieved by generating two 2D TFs based on two randomly drawn actions from $\mathcal{D}$ for the same scene. This comparison is conducted within a

preference interface in which both renderings of the associated actions can be visualized and compared. Here, human inspectors can choose the preferred visualization, resulting in $a_c \succ a_r$, where $a_c$ and $a_r$ represent the chosen and rejected actions. By following the Bradley-Terry model [10] for estimating score functions from pairwise preferences, the preference signal, which the reward model $r_\psi$ receives, is formulated as

$$p_\psi(a_c \succ a_r | s) = \frac{\exp(r_\psi(s, a_c))}{\exp(r_\psi(s, a_c)) + \exp(r_\psi(s, a_r))}$$
$$= \sigma(r_\psi(s, a_c) - r_\psi(s, a_r)), \tag{3}$$

where $\sigma$ is the sigmoid function. For the training of the reward model we decided to compare the loss function introduced by Christiano et. al. [7] with the loss function used by Ouyang et. al. [11], Bai et. al. [12] and others. The main difference between these two loss functions lies in the inclusion or exclusion of equal preferences. Including these equal preferences, the cross-entropy loss function is defined as

$$\mathcal{L}_C = -\mathbb{E}_\mathcal{D}\left[\mu_c \log p_\psi(a_c \succ a_r) + \mu_r \log p_\psi(a_r \succ a_c)\right], \tag{4}$$

where $\mu$ is a distribution over $\{c, r\}$ indicating which rendering the human inspector preferred. If both renderings are treated equally, $\mu$ is uniform. Treating the problem as a binary classification task yields the negative log-likelihood loss function

$$\mathcal{L}_L = -\mathbb{E}_\mathcal{D}\left[\log p_\psi(a_c \succ a_r)\right]. \tag{5}$$

## 3 Training

In this study, we utilized a proprietary dataset, which offers unique attributes to our domain of interest. Our complete dataset contains 16 CBCT head images with a size of $547 \times 421 \times 547$ and a pixel spacing of $0.2\,\text{mm}$. The individuals in the data set analyzed were primarily female, most of whom were in the 40-50 age range. The dental data are captured in the cranial region, extending from the chin to the zygomatic arch area. The data were acquired with a tube power of $83.7\,\text{kW}$ ($0.9\,\text{A} \cdot 93\,\text{kV}$) for an exposure time of $16.4\,\text{s}$ or $117.6\,\text{kW}$ ($1.2\,\text{A} \cdot 98\,\text{kV}$) for $11\,\text{s}$, respectively, and finally available in an anonymized DICOM format. Each image represents an individual scene for the RL framework. In the beginning, we tested our reward model only on one scene for which we collected 50 random actions and ten predefined actions based on a manual 2D TF design with a high-quality rendering result. From these ten pre-defined actions, an additional 40 actions were collected by data augmentations using small random shifts in the polygon vertices. From these actions, a

total of 4000 preferences were collected by one human observer, in which different renderings resulting from those actions were compared with each other based on a specifically designed priority list. Out of the 4000 preferences collected, a total of 613 are ambiguous, while 3387 are unambiguous.

The reward models are implemented in the PyTorch [13] framework. We trained the models for a total of 75 epochs on a batch size of 128, using the Adam optimizer [14] with the default learning rate. We observe that 75 epochs are sufficient for the training to converge. The training is repeated for each reward model three times, and we present the results from the runs with the highest value of $\mathcal{L}_C$ and $\mathcal{L}_L$. Although a higher loss at the end of a training would suggest worse performance, we found that these models achieve better results using our own evaluation methods. This behaviour was also previously discussed by Stiennon et al. [15].

## 4 Results

To evaluate our models, we calculated the distributions of the rewards for $20\,000$ uniformly distributed random actions on the scene and created corner plots for each vertex of the polygon. For each corner plot, only one vertex was modified, while the others were fixed based on a predefined polygon representing a high-quality rendering. These plots show the calculated reward of the trained model for each pixel position of the vertex in the JH. Ideally, the number of actions with a low reward should be very high and decrease significantly towards high rewards. Hence, the reward model would have learned to favor only the chosen actions and to punish rejected actions. Figure 2 shows the distributions of the rewards for the two loss functions $\mathcal{L}_C$ and $\mathcal{L}_L$ tested in this work. All distributions show a higher frequency towards the lower rewards. Nevertheless, a comparably high occurrence towards medium rewards can be observed for the negative log-likelihood function from equation 5, as illustrated in Figure 2(b). In contrast, a very sharp decline in the distribution of rewards can be observed in Figure 2(a) for the cross-entropy loss function. Figure 3 also confirms these results in the corner point plots. While a range for all four vertices can be identified for both loss functions where a high reward is obtained representing greater human alignment, the drop in rewards for the cross-entropy loss function in figure 3(a) is significantly higher. This means that the range of high rewards for the individual corner points is smaller, and more positions can be excluded in the resulting calculation of the 2D TF compared to the negative log-likelihood loss function in Figure 3(b). An example with the corresponding rendering results for the fourth corner point is shown in the last row of Figure 3. In this example, for the cross-entropy loss, it is clearly

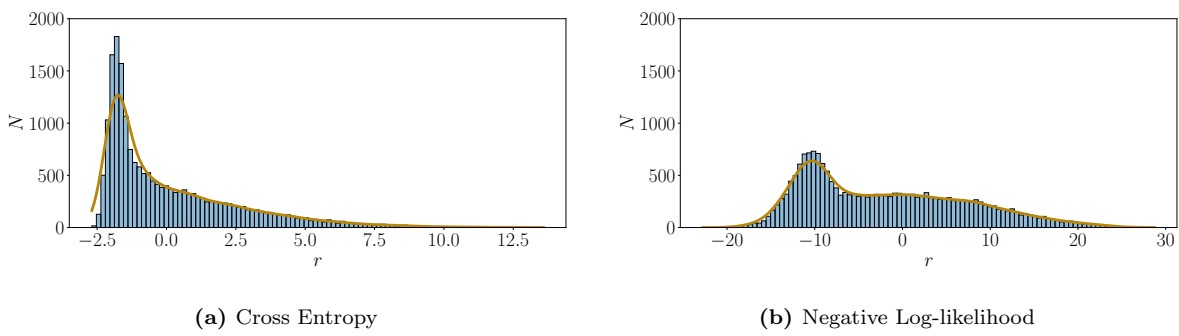

(a) Cross Entropy

(b) Negative Log-likelihood

**Figure 2.** Distribution of the calculated rewards by the trained reward model for both tested loss functions.

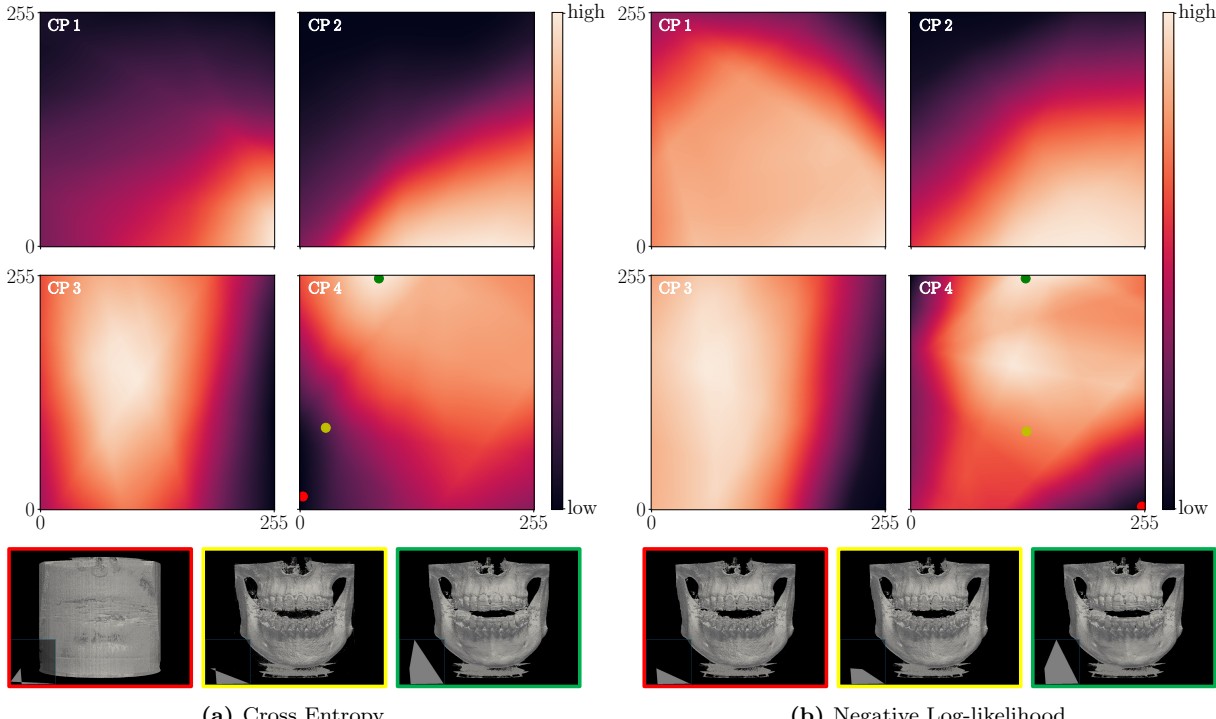

(a) Cross Entropy

(b) Negative Log-likelihood

**Figure 3.** Distribution of the calculated rewards by the trained reward model plotted for each pixel position in the 2D JH for each vertices of the polygon from the calculated 2D TF. For corner point four, the corresponding renderings are exemplary shown at three pixel positions for both loss functions.

shown that the pixel positions with a low reward also result in a worse visual rendering result. An improvement in the rendering results with increasing reward can also be observed for the negative log-likelihood function, even if the differences are less pronounced here and refer more to the dental area and the reduction of artifacts in this region.

Figure 4 shows the predicted rewards based on 12 additional labeled evaluation TFs for both loss functions applied in this work. In particular, very poor rendering results receive a very low reward for both loss functions, whereas the reward is typically higher for renderings with a clearer visual representation of the jaw. Despite the generally good assignments of the rewards, there are still individual outliers in both cases. For example, the rendering for the cross entropy loss in Figure 4(a) still receives a com-

paratively high reward in the second row and first column, despite some artifacts in the dental area. On the other hand, for the negative log-likelihood loss in Figure 4(b), particularly the still relatively high reward for the rendering of the third row and second column stands out. Here, hardly any anatomical structures are recognizable, so that the reward should be considerably lower.

## 5 Discussion

The results of the work already show that it is possible to train a reward model based on human feedback, which can later be used to train an RL agent to automatically generate 2D TFs for rendering results. However, the visual results from Figure 3 suggest

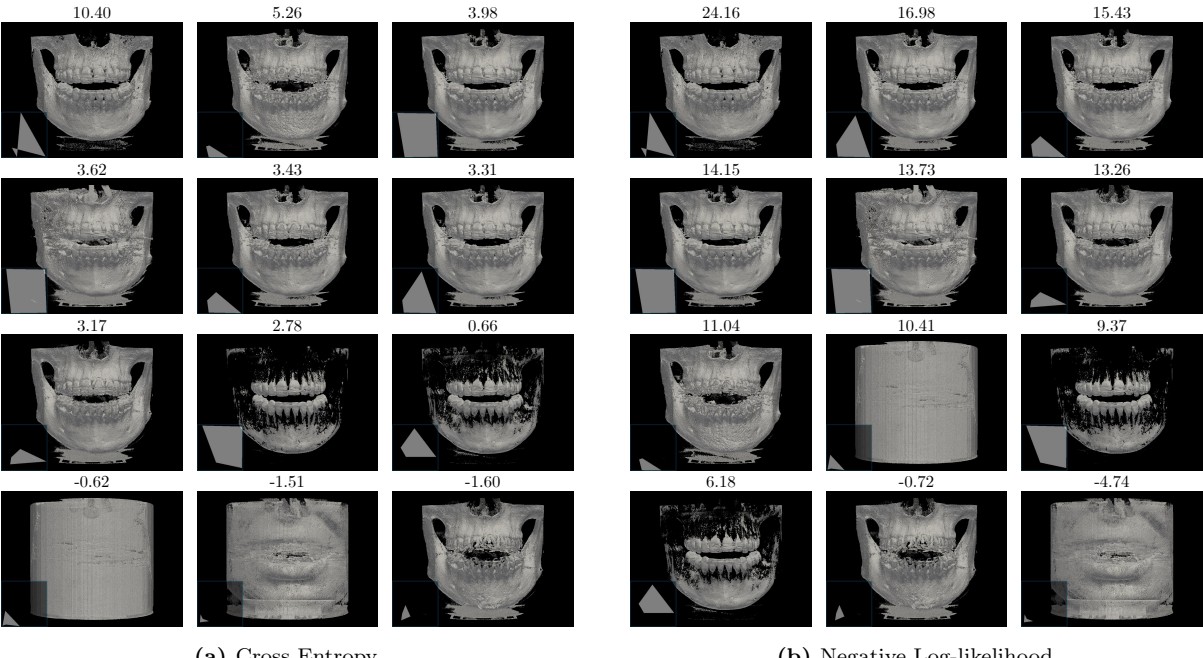

**(a)** Cross Entropy         **(b)** Negative Log-likelihood

**Figure 4.** Example results from the reward model with scores from highest to lowest from both investigated loss functions.

that an even more precise reward allocation should be applied for rendering results that are more similar to each other. The slightly better results of the cross entropy loss function could be due to the fact that the equal preferences are included in the evaluation, resulting in a better distribution of rewards for the current number of collected preferences. This effect could decrease with a higher number of collected preferences. Overall, the results could be improved if an even larger number of actions with a smaller deviation in the critical range of the JH were used to train the reward model. Consequently, more preferences could be collected, which would make the distinction between renderings even more accurate for the reward model. To identify this critical area, the already trained reward model can be used since, as shown in the corner point plot in Figure 3(a), it is already able to distinguish between areas with good and poor rendering results.

In addition to the action space per scene, the number of different scenes in total should also be increased so that the reward model does not overfit on one single example image. By increasing the number of scenes together with the corresponding actions per scene, however, the number of preferences to be collected would also rise accordingly, ensuring that the model has seen every combination per scene at least once if possible. Since preferences can also be collected from several users, creating an even higher level of objectivity, this effort would also be reduced per person. This is particularly advisable for pre-training, so that individualization should only take place when fine-tuning the model.

A further important aspect is to integrate an assignment of color values to specific anatomical regions in addition to the already assigned opacity. For this purpose, it could also be useful to integrate a pre-segmentation of certain anatomical structures, such as the teeth in our case, into the framework. Since this can already be generated fully automatically using AI models, this should not require a great amount of time in the processing pipeline.

# 6 Conclusion

With the development of a suitable reward model, which was trained from human feedback, a first important step has been taken towards the automated generation of a 2D TF for an optimized rendering adapted to the individual user.

In the future, we aim to train the reward model on even more scenes and corresponding actions, so that a RL agent can be developed, thus complementing the RLHF pipeline. Forthcoming work also involves evaluating the performance of the reward model using more complex feature extractors such as variational autoencoders [16] and vision transformers [17]. In addition to that, we aim to adapt and validate our method for other medical imaging systems, such as MRI and US, ensuring broader diagnostic tools.

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
