# OpenReview forum: "Human Aligned Reward Modeling for Automated Transfer Function Generation of 3D Rendering of Medical Image Data"
_NLDL.org/2025/Conference — Submitted to NLDL 2025_

### Official Review · Reviewer_H1t4 · 2024-10-02
**General Review**

**Confidence:** 4

**Summary:**

The work focuses on developing an advanced reward model to leverage RF studies. The concept is applied to 3D medical image rendering, where an automatic transfer function is proposed, reducing the dependence on human agents.

The authors tested their method on a proprietary dataset using a simple convolutional architecture and evaluating two loss functions. They visually inspected the rendered images to assess the trade-off between reward level and quality.

**Strengths:**

The paper suggests a way to automate the creation of transfer functions using reinforcement learning. This method can benefit the RF area and speed up other fields, such as the medical field.

Overall, the text is well-written, clear, and straightforward.

The images are self-explanatory.

The dataset is original.

**Weaknesses:**

It is unclear how this work compares to others of its kind. In other words, it would be interesting to benchmark it against similar approaches. Furthermore, are there other datasets to which the method could be applied to test its generalization capacity? The novelty should be compared with the state of the art. In addition, the following minor points deserve attention:


1 - I suggest shortening the title to: "Reward Modeling for 3D Medical Image Rendering Transfer Functions"

2 - In the abstract, there are abbreviations without the full terms.

3 - The authors compared different loss functions, but what is the rationale behind choosing the CNN architecture (2D or 3D)? Were others considered, such as Transformers?

4 - How were the hyperparameters adjusted? How was the data split for training, cross-validation, and testing?

5 - Is the dataset proprietary? Is it available? Is there a benchmark for this problem and dataset? How does the work compare to other studies of this type?

6 - The conclusion is vague. What worked and what went wrong? What lessons have been learned?

**Final Rebuttal Confidence:**

4

**Final Rebuttal Justification:**

Most of my comments have not been responded to. There is room for improvement and more studies should be done.

**Justification:**

Authors need to compare the method with the state of the art and test it on other datasets to attest to its generalization capacity.

---

### Official Review · Reviewer_YUYt · 2024-10-08

**Confidence:** 2

**Summary:**

This paper presents a new method that utilizes RLHF to help the 3D rendering of medical image data. The key idea is to learn an automatic transfer function generation model that is guided by human preference. Experiments demonstrate that the proposed method's promising performance.

**Strengths:**

- The idea of using RLHF to help the learning of TF generation is well-motivated and reasonable to me.
- The method is quite simple and easy to follow.
- The illustrations in the paper are helpful.

**Weaknesses:**

- There are no details on how the human inspectors produce the preference data.
- The evaluation is based on a few data samples, so it might not be very convincing that the method can be generalizable to any medical image data.
- The code is not released.

**Final Rebuttal Confidence:**

4

**Final Rebuttal Justification:**

I still have a major concern for the generalization of the method.

**Justification:**

See my comments in weakness.

---

### Official Review · Reviewer_7RXT · 2024-10-09
**The paper titled discusses an innovative approach to automate the design of transfer functions (TF) for direct volume rendering (DVR) of medical images like CT scans.**

**Confidence:** 4

**Summary:**

In this paper the authors focus on carrying out RLHF in a traditional sense for the medical image data. They learn a reward model from human feedback to address the complexities of manually designing transfer functions for direct volume rendering of medical images, such as CT scans. This learned reward model is then used to train a reinforcement learning agent, which is capable of automatically generating 2D transfer functions that meet visual expectations and requirements.

**Strengths:**

1) The paper is an interesting use case for applied research - i.e., the use of RLHF for automated generation of TFs which is novel. This approach can potentially reduce the
2) The authors detail the entire pipeline - such as the reward model training, preference collection - which shows clear description of the methods carried out. Figure 1 is also a great visual to show how the method integrates into standard RLHF paradigm
3) The paper also compares how the reward prediction is affected by different loss function i.e cross-entropy vs NLL loss which shows the reliability in results.

**Weaknesses:**

1) Its unclear if there are some sort of baselines that already tackle the TF generation process? The paper could be strengthened by including comparisons to those methods if some exist in literature.
2) Had some experiments been carried out for datasets other than the CBCT ones? to see if such results could be generalized to other CT scans (other than head scans) - this would be interesting as its an applied project (would show us the extent of practicality of the approach)
3) The authors seem to have a plan for future work i.e training the reward model on a broader range of scenes and actions, potentially incorporating more complex feature extractors like Variational Autoencoders or Vision Transformers to refine the model's ability to generalize across different medical imaging scenarios. These would make the work more robust (which could be one of the minor weaknesses in the current draft)

**Final Rebuttal Confidence:**

4

**Final Rebuttal Justification:**

Would have liked authors respond to the comments and provide more justifications to other review comments as well (which are questions I have now as well)

**Justification:**

Overall its an interesting work of applying RLHF for learning reward functions from preferences in medical imaging data such as CT scans and is an interesting addition as applied work in the domain.

---

### Meta-Review · Area_Chair_pJB1 · 2024-11-01

**Recommendation:** Reject
**Confidence:** 5

**Metareview:**

The reviewers agree that the problem of learning an RL-based reward model for a 2D transfer function is relevant. There is, however, also agreement that the paper does not sufficiently relate to state-of-the-art. There is a missing description of related work, there is no comparison to other methods, and the description of how the data labels were created is insufficient. Further, results are shown on one type of data, and therefore it is difficult to tell how the method would generalize. Finally, the concerns raised by the reviewers were not sufficiently addressed in the rebuttal. Since the reviewers agree on their concerns, the recommendation is to reject the paper.

**Suggested Changes To The Recommendation:**

2: I'm certain of the recommendation.  It should not be changed

---

### Decision · Program_Chairs · 2024-11-06

Reject